# Massive Loss of Transcription Factors Promotes the Initial Diversification of Placental Mammals

**DOI:** 10.3390/ijms23179720

**Published:** 2022-08-26

**Authors:** Xin-Wei Zhao, Jiaqi Wu, Hirohisa Kishino, Ling Chen

**Affiliations:** 1Guangzhou Institutes of Biomedicine and Health, Chinese Academy of Sciences, Guangzhou 510530, China; 2Graduate School of Agricultural and Life Sciences, The University of Tokyo, Tokyo 113-8657, Japan; 3Department of Molecular Life Science, Tokai University School of Medicine, Isehara 259-1193, Japan; 4The Research Institute of Evolutionary Biology, Yoga 2-4-28, Setagaya-ku, Tokyo 158-0098, Japan

**Keywords:** transcription factor, macroevolution, mammal, gene loss, molecular evolution

## Abstract

As one of the most successful group of organisms, mammals occupy a variety of niches on Earth as a result of macroevolution. Transcription factors (TFs), the fundamental regulators of gene expression, may also have evolved. To examine the relationship between TFs and mammalian macroevolution, we analyzed 140,821 de novo-identified TFs and their birth and death histories from 96 mammalian species. Gene tree vs. species tree reconciliation revealed that placental mammals experienced an upsurge in TF losses around 100 million years ago (Mya) and also near the Cretaceous–Paleogene boundary (K–Pg boundary, 66 Mya). Early Euarchontoglires, Laurasiatheria and marsupials appeared between 100 and 95 Mya and underwent initial diversification. The K-Pg boundary was associated with the massive extinction of dinosaurs, which lead to adaptive radiation of mammals. Surprisingly, TF loss decelerated, rather than accelerated, molecular evolutionary rates of their target genes. As the rate of molecular evolution is affected by the mutation rate, the proportion of neutral mutations and the population size, the decrease in molecular evolution may reflect increased functional constraints to survive target genes.

## 1. Introduction

Gene expression patterns vary among species—even those which are closely related and share highly similar genomic sequences. These differences are believed to be important factors in organism evolution, being a major source of among species phenotypic variation [1]. Transcription factors (TFs) are sequence-specific DNA-binding trans-regulatory proteins of gene expression that perform an initial step of DNA decoding [2]. Moreover, TFs have many other important functions in eukaryotes [3,4]. As revealed by laboratory experiments, the ability of TFs to drive phenotypic changes has long been known. For example, Homobox (HOX) TF genes play a key role in proper body pattern formation [5], while Sex Determining Region Y (SRY) TF gene, is important for sex determination [6,7]. In addition, the mutations of TF genes have many unexpected consequences, such as the formation of induced pluripotent stem (iPS) cells [8] and cancers [9,10].

Simplification and complication are both critical aspects of macroevolution. Simplification, that is, the reduction in biological complexity, has received less scientific attention than complexity. Examples of simplification-driven biodiversification across the tree of life include early evolutionary histories of metazoans, fungi and eukaryotes [11]. In contrast to nonadaptive simplification, such as drift, which can lead to the accumulation of slightly deleterious mutations in bacteria [12], adaptive genome reduction may also explain some important stages of eukaryotic evolution, such as the simplification of animal metabolism [13]. The ‘less-is-more principle’ suggests that the loss of gene function is a common evolutionary response of populations undergoing an environmental shift and, consequently, a change in the pattern of selective pressures [14]. In this regard, TF patterns may naturally evolve along with organismal macroevolution.

Available studies show that some of important components of metazoan and embryonic–plant TF kits were present much earlier in their respective single-cell ancestors [15]. Given that the origin and expansion of TFs occurred long before the big bang of speciation, macroevolution has likely been driven by a more direct factor, which is TF loss. For example, the convergent simplification of adaptin complexes in flagellar apparatus diversification caused major diversification in eukaryotes [11]. Another example is functional and phenotypic diversification of the animal mouths by the loss of signaling factors complex of the Wnt family, which evolved 650 million years ago (Mya) in the early multicellular animals [16].

So far, the exact mechanism of TFs, which work during the fast adaptation of mammals to changing environments via macroevolution, is poorly understood. Advances in comparative genomics have clearly shown that the exclusive use of genes as evolutionary units is an oversimplification of real evolutionary relationships [17,18]. Orthologous groups (OGs) refer to sets of genes that have descended from a single ancestral gene in a given ancestral species or taxonomic level. In this research, we use orthologous groups of TFs on the mammal level, rather than TF families to category TFs and construct phylogenetic trees. The OG category approach maintains the evolutionary events in mammalian history and avoids the difficulty of constructing trees for large TF families such as C2H2. By reconciling OG trees of TFs to mammal species tree, the gain and loss events of TFs can be detected. However, for trait values, only current species are available and certain. Here, we use the presence and absence of TFs on terminal branches to detect the possible association with trait divergence. In this study, we show the pattern of TF loss and address the role of TFs in the macroevolutionary process of mammals by describing the correlation between TF loss, target gene (TG) and molecular evolutionary rate, and also between TF loss and species traits. 

## 2. Results

### 2.1. The Analysis of Evolutionary History of Transcription Factors (TFs) in Mammalians

To analyze the evolutionary history of TF birth and death events, we de novo identified 140,821 TFs from 96 mammalian species [19] and grouped them into 1651 orthologous groups, each having a single origin at the root of mammals (Figure 1a). The history of TF duplication and loss in each orthologous group was estimated by reconciling the gene tree with the species tree (Figure 1b). 

The most recent common ancestor of extant mammals is dated back to approximately 170 million years ago (Mya) (Figure 1c). After splitting from Marsupialia, Placentalia diverged into Afrotheria, Xenarthra and Boreoeutheria about 100 Mya. We repeated the analysis by successively assuming Atlantogenata, Exafroplacentalia and Epitheria topologies at the same threshold level. The events detected under the three different topological arrangements were very similar.

Between 100 and 95 Mya, mammals, especially placental mammals underwent initial diversification, the average TF loss rate reached its first peak of 182.4/My (Figure 1c,d). The second peak of TF loss occurred near the Cretaceous–Paleogene (K–Pg) boundary, at 66 Mya. The TF loss rate was as high as 64.5/My within a 4-Mya window of the K–Pg boundary. 

As shown in Figure 1d, the most rapid TF gains mainly occurred during the early stages of mammalian macroevolution; this was especially the case for Boreoeutheria, with 1480 TF gain events at corresponding rates of 161.5/My. After this period, TF loss dominated. We detected 1967 TF loss events occurring at a rate of 284.7/My along the ancestral branch of Euarchontoglires. In addition, 2879 TF loss events at a rate of 345.0/My were inferred along the ancestral branch of the Laurasiatherian lineage, while 3819 taking place at a rate of 209.3/My were uncovered on the ancestral branch of marsupials. TF loss also occurred rapidly on subsequent branches, especially where common ancestors of different mammalian orders started to differentiate. For example, the estimated TF loss rates on Chiroptera, Perissodactyla and Carnivora ancestral branches were 279.6, 115.8 and 313.2/My, respectively. During this time period, the number of species increased significantly. Highly diversified lineages would be expected to contain more highly diversified TFs as well, but we found the opposite to be true. The TF pool became simplified during mammalian species diversification, indicating that TF lost may be related to the early diversification of mammals. 

For species radiations, the short branches and incomplete lineage sorting poses many challenges in resolving the tree of life (reviewed by [20]). The Nothung software which we used in this study, enabled the estimation of the inconsistency between gene tree and species tree, excluding the incomplete lineage sorting and short branches due to ancestral polymorphism [21]. For this purpose, four levels of effective population size (10^4^, 10^5^, 10^6^ and 10^7^) were applied as a threshold when considering incomplete lineage sorting and short branches (Figure 2). Although the number of events varied extensively among the four population-sized thresholds, the overall trend was quite similar in regard to TFs. As a representative example of the overall trend, but not exact numbers, the results obtained using an effective population size of 10^6^ are shown in Figure 1.

To avoid the possible variation due to the quality of the mammal root species, we traced 982 ancient mammalian TF orthologous groups, which originated before mammal common ancestor by inclusion an additional 11 outgroups (four birds, four reptiles, two amphibians and one coelacanth) and 82 mammalian species (Figure 3a and Appendix A). From approximately 170 to 100 Mya, the rate of ancient mammal TF gain was slightly higher than the rate of loss, and then after this period the loss of TF started to dominate then loss dominated (Figure 3b). 

### 2.2. Association with Trait Values

To examine the effect of TF loss on life history traits, we regressed four binary traits—sociality, diurnality, reproductive seasonality and insectivory—on the states (presence/absence) of 1-to-1 orthologous TFs via lasso logistic regression (Figure 4, Appendix A). 

TFs, especially members of the zinc-finger protein with Krüppel-associated box domain (KRAB-ZNF) family, are prominent candidates for a role in mammalian speciation [23]. Consistent with this idea, ZNFs or KRAB-ZNFs comprised half of the TFs significantly associated with the four life history traits (Appendix A).

As for sociality, 29 associated TF genes had non-zero correlation coefficient values. This result indicates that the loss of these TFs is possibly linked to the sociality trait. Highly social species have an increased risk of parasitic infection, which necessitates a heavy investment in immune functions [24]. Thus, among genes associated with sociality, half were related to immunity-associated phenotypes (Appendix A). Sociality has different effects on shape and behavior, including the relative size of the brain and the prevalence of infanticide (reviewed by [25]). De novo mutations in NACC1 and disruption in EPM2A gene can lead to cerebral atrophy and infantile epilepsy [26,27]. Epilepsy can have significant negative consequences on a child’s social development [28]. The ancestor of placental mammals experienced rapid changes in these traits around the K-Pg boundary period [29]. Consequently, TF loss is considered to help the preservation of solitariness in extant solitary mammals that is also reflected in our results (Table 1). 

In total, 24 TF genes were significantly associated with diurnality. Starvation and a cold environment have caused the transition from nocturnal to diurnal habits in mammals, and metabolic balance has been identified as a potential common factor affecting circadian rhythm organization [30]. Functions of TFs associated with diurnality shed light on their adaptation. Overexpressed ZNF667 (MIPU1) decreases oxLDL-induced cholesterol accumulation [31]. ZNF648 is associated with high-density lipoprotein cholesterol levels [32]. Knockout of the ZFP92 gene increases the amount of total body fat [33]. GBX1-knockout mice exhibit reduced thermosensory functions [34]. Changes in the metabolic balance of fat and thermosensory function might affect changes in tolerance to starvation and cold that likely promoted the adaptation to a nocturnal lifestyle. 

The type of food ingested by mammals is also evolutionarily very important. Among genes associated with the insectivory trait, the knockout of SEBOX, PHOX2A and KLF8 genes in mice has been found to increase the levels of circulating total protein or improve glucose tolerance [33,35]. Knockout of NSL1 and ZNF648 leads to increased circulating cholesterol levels [32,33]. According to various knockout data [33,36], AFF1, NSL1 and TULP1 are related to vision. Thus, TF loss likely played an important role in the divergence between insectivory (poor eyesight) and non-insectivorous (good eyesight) mammals.

In terms of reproductive seasonality, 23 TFs are significantly associated with this trait. More than half of the reproductive seasonality-associated TFs were ZNFs. KRAB-ZNFs play a major role in the recognition or transcriptional silencing of transposable elements, and transposon-mediated rewiring of gene regulatory networks that contributed to the evolution of pregnancy in mammals [37]. With regard to the phenotype associated with these TF genes (Appendix A), 13 out of 23 are related to fetal viability or formation. Seasonally breeding mammals with these TF losses most likely have better survival chances for fit to specific niches. 

### 2.3. The Effect of TF Loss on Its Current Surviving Target Genes

To quantify the effect of TF loss on transcription factor’s target genes (TF-TGs), we examined TG evolutionary rates based on 9396 TF-TG interactions by literature curations from the TTRUST2 database [38], and 1,342,129 TF-TG interactions based on Chip-seq data from the hTFtarget database [39] (Figure 5). Figure 5b,c, which compare the evolutionary rates of 1-to-1 orthologous TGs with TFs and the evolutionary rates of TGs without TFs, clearly show the decelerating effect of TF loss on TGs over the long term (paired *t*-test, *p*-value = 0.017 for TF-TG interactions by literature curations; *p*-value < $2.2\times 10^{-16}$ for TF-TG interactions based on Chip-seq data). If it is the decelerate evolutionary rate of TG which drives the TF loss, we will see many low-rate TGs with TFs, or many points on the left side of Figure 5b,c, but they do not show this pattern. Because the rate of molecular evolution of a gene is negatively correlated with the strength of a functional constraint [40], the decelerated gene tends to be conserved. As the target genes covers more than fifteen thousand genes, it suggests that currently surviving genes have been “protected” by TF loss over a long time period, especially during macroevolution. 

In this study, we have uncovered three main findings. First, the number of TFs increased greatly during the early stages of mammalian species formation; later, however, TF losses predominated during the course of macroevolution. Second, TFs significantly associated with traits change. Third, in the face of serious environmental changes, such as the K–Pg boundary, the loss of such TFs rewired the regulatory network, set functional constraints on TGs and facilitated organismal survival. For TFs, duplication provides possibility, but loss produces inter-species differentiation.

## 3. Discussion

Our study showed that the main branches of mammals, especially placental mammals, Afrotheria, Xenarthra and Boreoeutheria experienced a large number of TF loss events at approximately 100 Mya, which corresponds to the first peak of TF loss in mammalian evolutionary history (Figure 1). The likely place of origin or current extant range of Afrotheria, Xenarthra and Boreoeutheria clades is Africa, South America and the northern supercontinent of Laurasia, respectively [41,42]. This distribution indicates that the living environment of these organisms has likely changed compared with their common ancestor; in other words, their niches and eco-traits may have changed. In contrast, a significant correlation exists between traits and the loss of some of the TFs (Figure 4). This correlation suggests that TF loss may be beneficial to the survival of species undergoing niche and environmental changes and can therefore become fixed in the survivors. At the second peak of TF loss, which took place near the K–Pg boundary (66 Mya), clusters of loss events are especially apparent on three branches leading to Carnivora, Perissodactyla and Cetartiodactyla. Members of Carnivora are basically carnivores, while Perissodactyla mainly comprises herbivores and Cetartiodactyla contains both herbivores and omnivores. The feeding preferences of the three taxa have thus clearly and rapidly diverged from those of their common ancestor, and the massive loss of TFs in this transition may also have been beneficial to species survival. This phenomenon suggests that TF loss may play an important role in species adaptation to environmental change and macroevolution.

The loss of genes is generally thought to lead to non-functionalization [43]. These genes are usually considered to be redundant parts that can be replaced by other genes or provide materials for evolution. While TF loss leads to different results. A positive correlation exists between the loss of TFs and mammalian ancestral traits (Figure 4). Losses of TFs decelerate the molecular evolution of TGs (Figure 5). The loss of TFs in mammals enhances functional constraints on their TGs. TF loss may keep target genes with fitness privileged.

Mutations tend to have functional loss than gain of function [44]. For TFs, this may explain why TF loss can be a rapid response to change in the environment rather than TF gain. Research on bacterial populations under different conditions has also shown that adaptive loss-of-function by the mutations of enzymatic and regulatory functions has an important role in their adaptation to new environments [45]. Possibly the “Ratchet effect” on regulatory system redundancy to improve fitness during environmental challenges. According to our previous research, TFs are located on a more peripheral state on the gene interaction network than other functional genes; thus, it is more dispensable and milder for species to endure changes [19]. This may provide clues on why TF loss takes place prior to other functional gene variations. Facing environmental changes, TF loss may increase fitness by adjusting more privileged traits directly and preserve fitness genes by functional constraints. 

In this study, although TF loss is generally predicted to be one of the factors that may affect its target genes, macroevolution and traits, the opposite direction is also possible. However, if the loss of TF is driven by the low evolutionary rates of its target genes, differentiation and trait changes, it could also demonstrate that the loss of TFs could privilege species that survive these changes.

Many TFs have been lost during mammalian history, especially around the K–Pg boundary or afterwards. The massive extinction of dinosaurs as predators may have reduced functional constraints on relevant transcription machineries in our ancestors. As a result, TFs were lost unless new species shifted their habitats into recently opened niches and developed new lifestyles adapted to the new environments. However, TF loss has enhanced functional constraints on any TF TGs surviving to the present day. Mammals that lost these TFs may have increased their survival rate by retaining the ancestral traits in an unchanged niche. When an old niche was occupied, a less-adapted species with a less-altered or insufficiently altered transcriptional machinery may have needed to take risks to adapt to a new niche and change traits for survival. To adapt to new environment, a new set of TF loss occurred. With different TF loss patterns, species in old niches and in new niches owned different traits. Their phenotypes became different from their ancestor or each other in a relatively short period of time. During macroevolution periods, ancestor species in different niches may keep different TF profiles and gene profiles due to TF loses, thus promoting biodiversity.

## 4. Materials and Methods

### 4.1. Mammalian Transcription Factors (TFs)

Based on our previous research, we obtained 140,821 TF proteins, nearly all of which were specific for mammals [19]. To assess the accuracy of TF annotations, we compared our list of 1625 human TFs (extracted from the 140,821 TFs in 96 mammalian genomes shown in Data Availability) with two well-known TF databases, AnimalTFDB3 [45] and HumanTFs [4] (Appendix A). The number of human TFs in these two databases is similar to that of our list: 1639 in humanTFs and 1665 in AnimalTFDB3. A total of 1402 TFs are listed in all three databases, while 82 are unique to HumanTFs, 123 are only found in AnimalTFDB3, and 140 are restricted to our human TF list. All three databases are based on similar DBD and HMMER [46] pipelines. 

AnimalTFDB3 contains 125,135 TFs from 97 genomes ranging from *Caenorhabditis elegans* to mammalian species such as humans, whereas our database contains 140,821 TFs and focuses on 96 mammalian species. The latter database may thus provide better insights into the evolutionary history of TFs in mammals. To avoid the limitations of mammalian species, we further annotated our database with information on orthologous groups from OrthoDB [47]. This way, we can trace TFs in mammalian species back to those of bacterial species.

To further confirm events occurring in the common ancestor of mammals, we conducted a hidden Markov model (HMM) search on eleven outgroup species (four birds, four reptiles, two amphibians and one coelacanth) and collected all genes predicted to encode TFs. 

### 4.2. Multiple Alignment of Mammalian Orthologs 

We downloaded 96 complete mammalian genomes from GenBank [48] and used a custom Perl script to extract protein-coding sequences of each species. A gene pool of 21,350 mammalian genes was constructed based on NCBI genomic annotations. Using the results of the HMM search and annotations in OrthoDB, we assigned TFs to 2880 mammalian orthologous groups. We confirmed the homology of outgroup TFs and mammal-TFs by a local blast search (blastp, [49]) and assigned outgroup TFs to mammalian orthologous groups.

We generated a multi-sequence file of all genes in the gene pool for each of the orthologous groups. Codon-level alignments were performed in Prank v.170427 [50]. Sites with less than 70% coverage across all species, as well as sequences with less than 30% coverage among gene loci, were removed from the alignment. 

### 4.3. Inference of Gene Trees 

We estimated the maximum likelihood tree for each gene using the IQ-TREE software [51], which automatically performed model selection and determined the best data partitions. The best evolutionary model for each gene was independently selected based on the Bayesian information criterion and used for the inference of the nucleotide tree. All gene trees were calculated using 1000 bootstrap replicates.

### 4.4. Species Tree Inference and Divergence Time Estimation

We selected 823 genes that were single copies in mammals according to Wu et al. [29] and present in all 96 mammals. The 823 genes were used to infer a species tree using the coalescent-based Njst method [52]. The topology of the inferred species tree was consistent with that of Tarver et al., 2016 [53], who placed treeshrew (*Tupaia chinensis*) as the root lineage of Glires. The phylogenetic position of treeshrew is not yet resolved; however, several researchers consider treeshrew to be the root lineage of Euarchonta not Glires [54]. As an alternative species tree in this study, we consequently used a tree in which the position of treeshrew was fixed at the root of Euarchonta rather than Glires. Following the same method as Wu et al., 2017 [29], we estimated the divergence times of 96 mammals based on the inferred branch effect (genomic rate × genomic time) and fossil calibrations.

### 4.5. Reconstructing the History of TF Duplication and Loss

A history of gene duplication and loss was estimated by reconciling gene trees with the species tree and plotted by R. To focus on these events during mammalian evolution, TFs were categorized into orthologous groups (OrthoDB, https://www.orthodb.org/, accessed on 18 December 2019), each having a single ancestor at the root of mammals. Out of 140,821 TFs from 96 mammalian species, OrthoDB included nearly 120,000 TFs from 82 species. These TFs were assigned to 2880 orthologous groups. A phylogenetic tree was constructed for each of the 1651 orthologous groups containing more than three members. These 1651 OGs cover over 97% TFs of 82 mammal species. To infer the history of TF duplication and loss, each of the 1651 phylogenetic trees was reconciled with the species tree according to the criterion of parsimony using NOTUNG 2.9 [21]. To take into account inconsistency due to incomplete lineage sorting, we established a branch length threshold in terms of Nμ. Branches with lengths under the threshold were treated as weak branches and rearranged by NOTUNG. For each orthologous group tree, the value of μ corresponded to the average path length from root to terminals divided by the calculated age of the root. We assumed an average generation length of 10 years and considered four possible values for effective population size, namely, 10^4^, 10^5^, 10^6^ and 10^7^, and three tree topologies (Appendix A). The events of the orthologous groups were aggregated to obtain the numbers of births and losses of TFs along each branch of the species tree. In this paper, we present the results of the analysis based on an effective population size of 10^6^. Other effective population size threshold settings gave similar results. The platypus genome [55] is not updated on the current version of OrthoDB. To confirm that the inclusion of this species did not lead to biased results, we added 11 more outgroups with 82 mammalian species, including 1 fish, 2 amphibians, 4 reptiles and 4 birds, for use in tree construction. Using the 11 outgroups, 982 TF orthologous groups were traced and analyzed by the above-mentioned method. Divergence times of the outgroups were taken from the Timetree [22] database. 

### 4.6. The Effect of TF Loss on TG Molecular Evolutionary Rates

To examine the long-term effects of TF loss, we compared the mean rates along the terminal branches of a TG tree between species that possess the TF and species that do not. Rates of molecular evolution were estimated as the ratios of branch lengths to the relevant time periods in the species tree. In total, 9396 TF-TG interactions by literature curations were downloaded from the TTRUST2 database [38] and 1,342,129 TF-TG interactions based on Chip-seq data were downloaded from the hTFtarget database [39].

### 4.7. Association Study of Ecotype Traits and TF Presence/Absence

We manually collected the ecotype traits of 96 mammalian species from the Animal Diversity Web (http://animaldiversity.org/, accessed on 15 August 2019). The presence and absence of TF orthologs in each species were recorded as 1 and 0, respectively, based on mammal-terminal single-copy orthologs. The ecotypes were binary recorded as 1 and 0. We identified significant associations using the lasso logistic regression procedure as implemented in the glmnet package [56] in R.

## Figures and Tables

**Figure 1 ijms-23-09720-f001:**
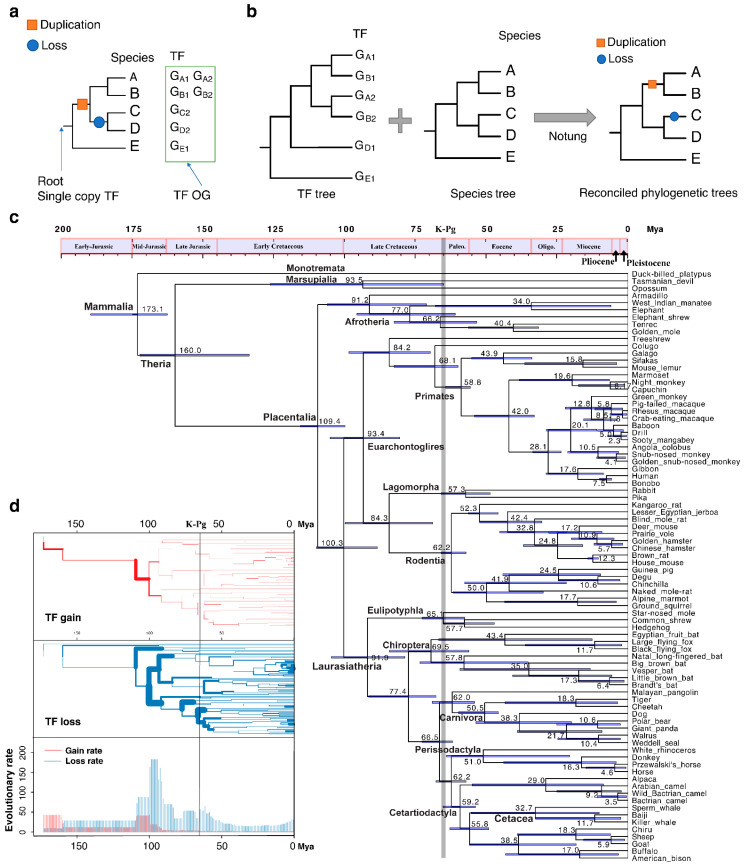
Simplification of transcription factors (TFs) during mammalian evolution using an effective population size of 10^6^. (**a**) A simple example of TF orthologous group (OG). The orange square indicates a duplication event, and the blue circle represents a loss. The genes in the green box belong to the same OG and all originated from the single-copy gene G at the root. (**b**) Inference of duplication and loss by TF-tree–species-tree reconciliation. G, TF genes. A–E, species. (**c**) Time tree of 82 mammals based on protein sequences. Numbers at internal nodes are estimated divergence times (Mya, million years ago), with 95% credible intervals indicated by horizontal bars spanning nodes. (**d**) Atlas of TF gain and loss during mammalian history. The width of each branch is proportional to the rate of TF gain (or loss). The bar plots are mean evolutionary rate (mean TF gain or loss events of each million years).

**Figure 2 ijms-23-09720-f002:**
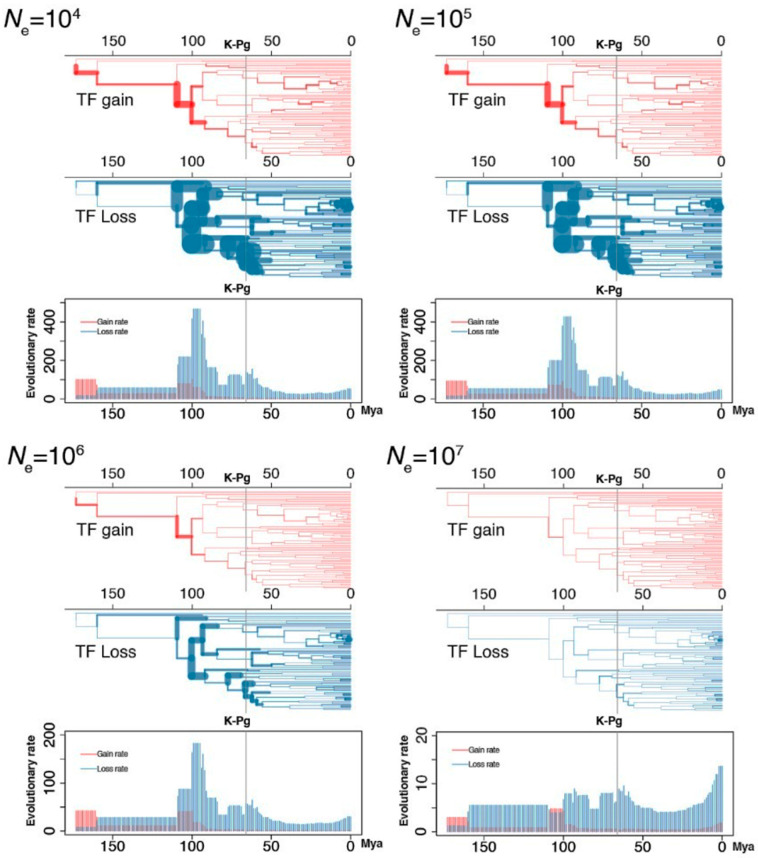
Transcription factor (TF) events inferred using four different effective population sizes (*N*_e_)—10^4^, 10^5^, 10^6^ and 10^7^—as a threshold when considering incomplete lineage sorting and short branches. TF losses and gains are represented by blue and red bars, respectively. The width of each branch is proportional to the rate of TF gain (or loss). The bar plots are mean evolutionary rate (mean TF gain or loss events of each million years).

**Figure 3 ijms-23-09720-f003:**
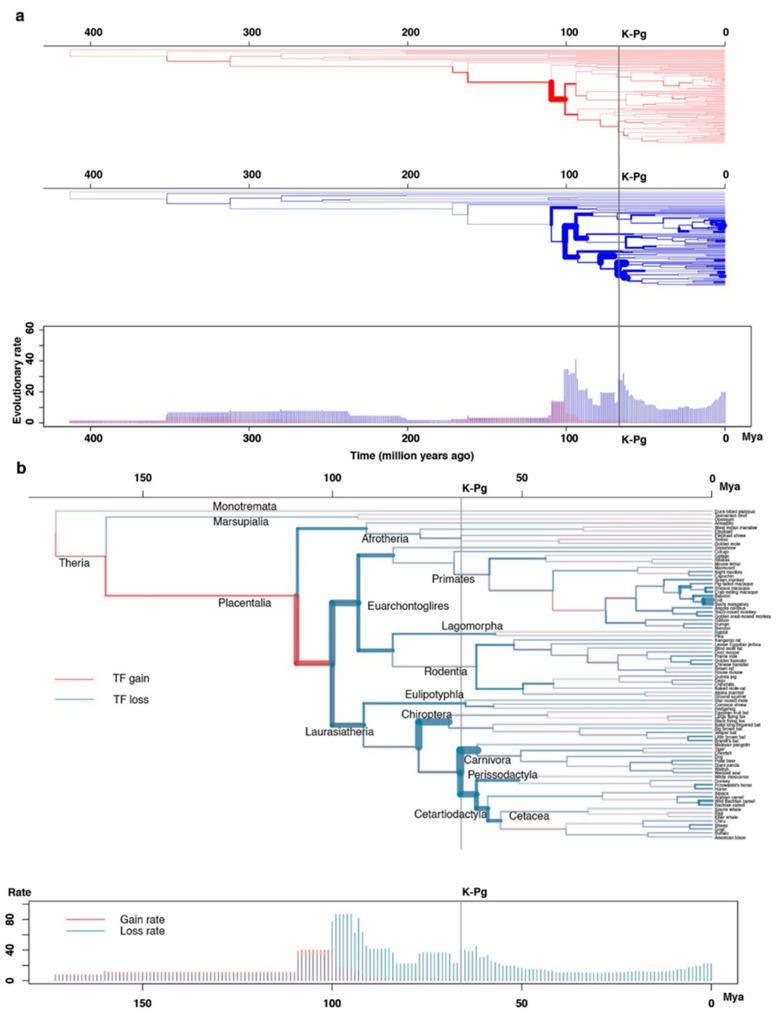
Atlas of ancient transcription factor (TF) gain and loss. (**a**) TF events in Atlantogenata inferred using an effective population size of 10^6^ with outgroup species included. TF losses and gains are represented by blue and red bars, respectively. The width of each branch is proportional to the rate of TF gain (or loss). Divergence times of outgroups (Aves, reptiles, amphibians, etc., see Appendix A) originated from Timetree [22]. (**b**) Atlas of ancient TF gain and loss of 982 orthologous groups of TFs during mammalian history. The width of each branch is proportional to the rate of TF gain (or loss). Time scale is million years ago (Mya).

**Figure 4 ijms-23-09720-f004:**
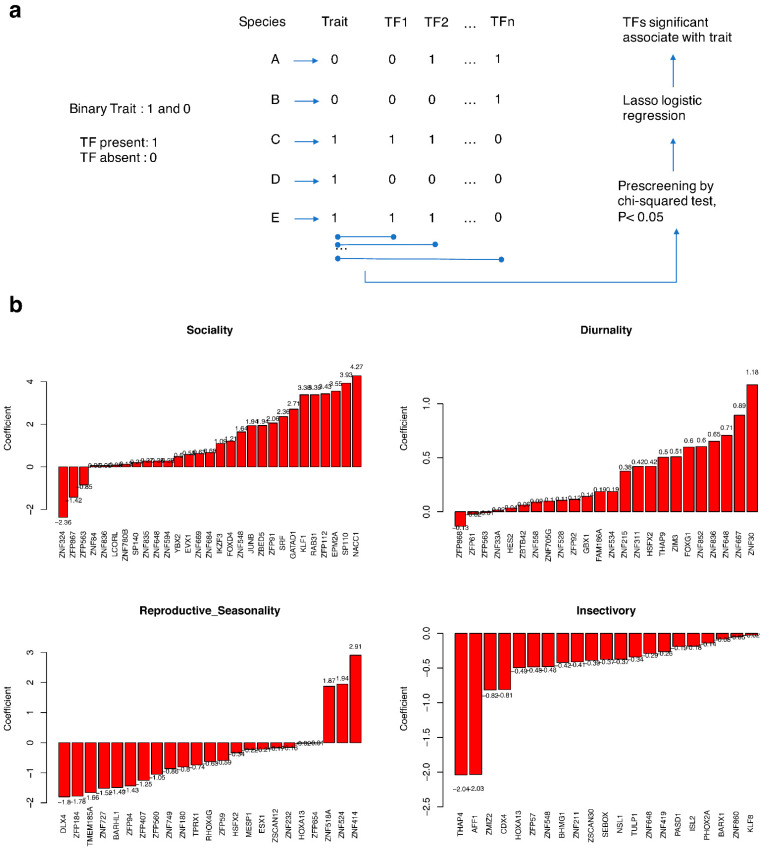
Association of TF loss with mammalian traits. (**a**) The pipeline used to detect an association between TF presence/absence and life history traits by lasso logistic regression. (**b**) TFs significantly associated with traits. The *x*-axis shows TF genes significantly related to the indicated trait, and *y*-axis values are lasso logistic regression coefficients pre-screened with a chi-square test (*p* < 0.05).

**Figure 5 ijms-23-09720-f005:**
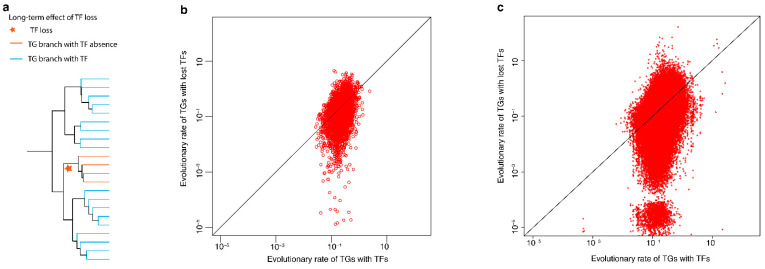
Deceleratory effect of the loss of transcription factors (TFs) on molecular evolutionary rates of their target genes (TGs). With decreased molecular evolutionary rate, current genes survive till present. (**a**) Comparison of the evolutionary rates of TGs in species in the subtree of the TF event (colored in red) with the rates of TGs in other species. (**b**,**c**) Long-term effect of TF loss on the evolutionary rate of TGs, TF-TG interactions by (**b**) literature curations or (**c**) based on Chip-seq data. *x*-axis: evolutionary rate of TGs still controlled by TFs; *y*-axis: evolutionary rate of TGs with lost TFs. TF-TG interactions by literature curations from TTRUST2 database; TF-TG interactions based on Chip-seq data from the hTFtarget database.

**Table 1 ijms-23-09720-t001:** TF loss in NACC1, SP110 and EPM2A associated with solitary.

Common_Names	Sociality	NACC1	SP110	EPM2A
Cheetah	solitary	0	1	1
Weddell_seal	solitary	0	1	1
Tiger	solitary	0	1	1
Duck-billed_platypus	solitary	1	0	1
Lesser_Egyptian_jerboa	solitary	1	0	1
Opossum	solitary	1	0	1
Galago	solitary	1	0	1
Orangutan	solitary	1	0	1
Tasmanian_devil	solitary	1	0	1
West_Indian_manatee	solitary	1	0	1
Hedgehog	solitary	1	1	0
Pika	solitary	1	1	0
Deer_mouse	solitary	1	1	0
Other 14 species	solitary	1	1	1
Elephant_shrew	social	1	0	1
Bottlenosed_dolphin	social	1	0	1
Other 67 species	social	1	1	1

TF absence and presence are represented by 0 and 1, respectively.

## Data Availability

TF data, program codes, phylogenetic trees and other data related to our research: (https://drive.google.com/drive/folders/19jvuchCudXDZyrcJza4CpsNf22enYkHd?usp=sharing) (accessed on 18 August 2022).

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
