# Peer review of "Massive Loss of Transcription Factors Promotes the Initial Diversification of Placental Mammals"

_ijms, 2022, doi:10.3390/ijms23179720_

Round 1

Reviewer 1 Report

This is an interesting study that speculates on the interplay of the loss/gain of transcription factors and evolutionary rates in mammals' macroevolution. The analyses are comprehensive and thorough, and the correlation observed is intriguing. There are, however, two concerns:

(1) As we all know, correlation does not imply causation. While TF loss might indeed be correlated with lower evolutionary rates, divergence, and four "life history traits", it is unclear if there are any directional causalities. Therefore, the authors' conclusions might be unwarranted --- or, at least, should be toned down in the Discussion.

(2) The Introduction section is not well-written. Specifically:

"K–Pg boundary" in the Introduction should be spelled out.

"massive TF losses are significantly correlated four life history traits" is unclear. What, exactly, is correlated with what? Why these four traits?

"As analysis 23 covers 1,342,129 TF-TGs by chip-seq data, 9,396 by literature curations and showed consistent result" --- this sentence is difficult to understand. What, exactly, "showed consistent result"?

That being said, the main result --- "TF loss decelerated, rather than accelerated, molecular evolutionary rates of their target genes, suggesting increased functional constraints to survive target genes" is sufficiently noteworthy in itself to be published. However, the authors should acknowledge that there are other factors that might affect evolutionary rates and that the evidence for direct TF loss -> lower evolutionary rates is circumstantial. 

Author Response

Comments and Suggestions for Authors

This is an interesting study that speculates on the interplay of the loss/gain of transcription factors and evolutionary rates in mammals' macroevolution. The analyses are comprehensive and thorough, and the correlation observed is intriguing. There are, however, two concerns:

(1) As we all know, correlation does not imply causation. While TF loss might indeed be correlated with lower evolutionary rates, divergence, and four "life history traits", it is unclear if there are any directional causalities. Therefore, the authors' conclusions might be unwarranted --- or, at least, should be toned down in the Discussion.

Response 1: We sincerely appreciate the valuable comments and completely agree with the reviewer. So, we added these descriptions in the Discussion: “In this study, although TF loss is generally predicted to be one of the factors that may affect its target genes, macroevolution and traits, the opposite direction is also possible. However, if the loss of TF is driven by low evolutionary rates of its target genes, differentiation, and trait changes, it could also demonstrate that the loss of TFs could privilege species that survive these changes.” (lines 234-238). We also revised the Discussion section, to make our conclusion tone down.

(2) The Introduction section is not well-written. Specifically: "K–Pg boundary" in the Introduction should be spelled out.

Response 2: Thank you for the valuable comment. We added the full name of “K–Pg boundary”at line 18.

"massive TF losses are significantly correlated four life history traits" is unclear. What, exactly, is correlated with what? Why these four traits?

Response 3: We apologize for the unclear description. In this work, we found that the massive TF losses are significantly correlated with sociality, diurnality, reproductive seasonality, and insectivory traits. We detected their divergence during mammal history in our previous research (Wu et al., 2017). To provide a clear abstract, we removed this sentence and described in Results. (lines 129-131)

"As analysis 23 covers 1,342,129 TF-TGs by chip-seq data, 9,396 by literature curations and showed consistent result" --- this sentence is difficult to understand. What, exactly, "showed consistent result"?

Response 4: We apologize for the confusion and removed this sentence from abstract. (line 24). The detail information of the TF-TGs that is use in this study, is summarized in Method at section 4.6.

That being said, the main result --- "TF loss decelerated, rather than accelerated, molecular evolutionary rates of their target genes, suggesting increased functional constraints to survive target genes" is sufficiently noteworthy in itself to be published. However, the authors should acknowledge that there are other factors that might affect evolutionary rates and that the evidence for direct TF loss -> lower evolutionary rates is circumstantial.

Response 5: Thank you for your valuable comment. As we wrote in the manuscript, “the rate of molecular evolution is affected by the mutation rate, the proportion of neutral mutations, and the population size”(Lines 22-23). These are also important factors to impact the evolutionary rates.

Reviewer 2 Report

Dear Editors,

Dear Authors,

The reviewed study entitled: “Massive loss of transcription factors promotes initial diversification of placental mammals” represents valuable and above all very interesting insight to the genetic based mechanisms of mammalian macroevolution. The applied methods are suitable and correct. The manuscript is quite good quality considering language presentation and substantive content; however, some editorial improvements are required, and some questions must be answered.

In conclusion, I recommend the reviewed manuscript for possible publication in the International Journal of Molecular Sciences periodical. All remarks, questions and fixes were placed in the attached pdf file (yellow highlights contain fixes and sentence suggestions, while red highlights contain comments and questions).

Thank you for another interesting manuscript that I could review!

Author Response

Dear Authors,

The reviewed study entitled: “Massive loss of transcription factors promotes initial diversification of placental mammals” represents valuable and above all very interesting insight to the genetic based mechanisms of mammalian macroevolution. The applied methods are suitable and correct. The manuscript is quite good quality considering language presentation and substantive content; however, some editorial improvements are required, and some questions must be answered.

In conclusion, I recommend the reviewed manuscript for possible publication in the International Journal of Molecular Sciences periodical. All remarks, questions and fixes were placed in the attached pdf file (yellow highlights contain fixes and sentence suggestions, while red highlights contain comments and questions).

Thank you for another interesting manuscript that I could review!

Response: We sincerely appreciate these very detailed and valuable comments. Based on these suggestions and comments, we hope that this manuscript will provide a better understanding for our readers. Below is a point-by-point response to the red highlighted and yellow highlighted comments.

Red highlights:

Point 1: “From approximately 100 million years ago to the present, losses dominated TF events without a significant change in TF gains.” (lines 18-19). You said the same in the sentence from lines 16-17. Please merge both sentences or just delete this one.

Response 1: We deleted this sentence based on your comment above. (lines 16-18)

Point 2: “Introduction” (line 31). I think short information what TFs are should be placed in this chapter.

Response 2: Thank you for your valuable comment. We placed the definition of TFs” Transcription factors (TFs) are sequence-specific DNA-binding trans-regulatory proteins of gene expression that perform an initial step of DNA decoding [2].”(lines 33-34)

Point 3: (line 36) Please delete enter and merge paragraph 1 with 2. Both of them include the same kind of information. You do not change the subject so no need to create new paragraph.

Response 3: Thank you for your valuable comment. We followed this comment and merged paragraph 1 with 2. (line 34)

Point 4: “HOX”, “SRY” (line 39). please expand this abbreviation.

Response 4: We apologize for the missing of full gene names. We added the full gene name “Homobox (HOX)” and “Sex Determining Region Y (SRY)”. (lines 37-38)

Point 5: (lines 68-75) This part of text MUST be explained better because it is hard to follow and, in my opinion, it does not correspond to the rest of paragraph. Especially the term "orthologous groups" is a new information here that is very imprecise. What is the concept of orthologous groups? "We used orthologus groups" of what TFs, genes or something else? Please clarify this and show the connection with the above and below sentences. I do not see how you studied the gain and loss events of TF and association between TF loss ans traits.

Response 5: We apologize for the unclear description. Orthologous groups (OGs) refers to sets of genes that have descended from a single ancestral gene in a given ancestral species or taxonomic level. In this study, we use orthologous groups of TFs on mammal level, rather than TF families to category TFs and construct phylogenetic trees. The OG category approach maintains the evolutionary events in mammalian history and avoids the difficulty of constructing trees for large TF families such as C2H2. By reconciling OG trees of TFs to mammal species tree, the gain and loss events of TF can be detected. But for trait values, only current species are available and certain. So we use presence and absence of TFs on terminal branches to detect the possible association with trait divergence by lasso logistic regression. (lines 64-73)

Point 6: (lines 78-81) In fact, there is no addressed specific aim of this study.

Response 6: We apologize for the not concise description and removed this sentence.(line 76)

Point 7: (lines 102, figure 1) The description in the figure 1a is misleading. In the lines 84-85 you wrote that all identified TFs were grouped in 1651 orthologous groups. So on the figure should be not G (gene) but TF and not gene tree but TF tree.

Response 7: We apologize for the unclear description. To provide more precise description, we changed “Gene” to “TF” in figure 1 and descriptions. (lines 496-502,Figure 1)

Point 8: “highly diversified genes” (lines 122) TFs or genes?

Response 8: We apologize for the confusion and revised it to TFs.(lines 106)

Point 9: “an effective population size of 106 (lines 133) Please include this information in the caption of figure 1.

Response 9: Thank you for the valuable comment. We added this information in the caption of figure 1. (lines 496)

Point 10: (lines 179-183) What do you mean here? It is unclear. What are you trying to say? What is the connection with the above information? Please clarify.

Response 10: We apologize for the unclear description. We tried to detect why the unique TF pattern of Elephant_shrew and Bottlenosed_dolphin happens compared with the remains social species. However, it may confuse the readers, so we removed this part. (line 147)

Point 11: (lines 193-194) You said the same in the line 184-185. Please unify.

Response 11: We apologize for the redundant description and removed this sentence. (line 157)

Point 12: (line 226) My proposition for the description of Y-axis: "Evolutionary rate of TGs with lost TFs".

Response 12: We revised this sentence based on your comment above. The description revised from “Evolutionary rate of TG with TF loss” to “Evolutionary rate of TGs with lost TFs”. (line 527. Figure 5)

Point 13: (line 234) All this paragraph is conclusions and should be placed at the end of the discussion chapter or as seperate sub-chapter entitled "conclusions".

Response 13: Thank you for your valuable comment. We placed these sentences to discussion part.(lines 234-238)

Point 14: (lines 243-251) You said already about this and in fact it belongs to Material and methods chapter.

Response 14: We apologize for the not concise description and removed this sentence. (line 197)

Point 15: (lines 275-276) The loss of TFs in mammals enhances or reduces the functional constrains of TGs?

Response 15: The loss of TFs in mammals enhances the functional constrains of TGs. (220-221)

Point 16: (lines 339-341) Interesting software, may I see input file of selected gene and resulted tree? How to construct such tree by this software?

Response 16: We attached the tree files and gene list at link of Data Availability. Sequences can download by NCBI database. The iq-tree could infer a ML tree from an alignment file with the best-fit model automatically selected through ModelFinder by –s function. (lines 292-296, 368)

Point 17: “single-copy”(line 345) Do you mean orthologs?

Response 17: OK, it means 1-to-1 orthologous here. (lines 299). We re-phased the term “single-copy”here.

Point 18: “Njst method [51].”(line 347) Is this some software for construction of another tree? Is it NJ tree? How was the procedure of species tree constructing? 823 genes were aligned (according to amono acid or nucleotide seqence?) and then used for topology inference?

Response 18: Thank you for your valuable comments. Here, Njst is a coalescent-method based species tree reconstruction method. It is a sister method to MP-EST. Both MP-EST and Njst method takes gene trees as input. By considering of the differences of gene tree topology and its coalescent time (here is the branch length information of each gene tree), they summarize the best species tree, which could both explain gene tree topology and inconsistency between gene tree and species tree due to ancestral polymorphism. The differences between MP-EST and Njst is that MP-EST takes rooted gene trees; while Njst method could take un-rooted gene trees as input trees.

We had aligned the nucleotide sequences of 823 genes at codon level using Prank v.170427. Each gene was aligned independently, and each gene tree were calculated by IQTree software with 1,000 bootstrap replicates. The gene trees are used as input trees by Njst, and a coalescent species tree were obtained and used in this study. We wrote the approach carefully in Method section in our paper. (line 302)

Point 19: “Following the same method as Wu et al. 2017 [28], we estimated the divergence times of 96 mammals based on the inferred branch effect (genomic rate × genomic time) and fossil calibrations.”(line 352-354) Can you describe closer the procedure of divergenece time estimation? Is still based on the 823 genes mentioned above? How did you generated the time scale tree? Willingly I would see input data and output raw tree. What calibration nodes were used?

Response 19: Thank you for your valuable comments. The method which we calculated the time tree is our new method, with the detail of this method written in Wu et al. 2017 at section “Common and specific rates of protein evolution”. We think there are 3 factors that impact the variations of branch lengths of gene trees: branch effect, gene effect and gene-branch interactions. Here is our modelling. Branch effect is genomic mutation rate multiple time duration. It can be obtained by applying multiplicative ANOVA to the inferred branch lengths of the gene trees. In Wu et al. 2017, we further proved that the branch-effect based time tree estimation method is insensitive to the choice of data (DNA, or protein or codon) or method (from JC69 to codon method, or protein methods, Figure 1 and S1 in Wu et al., 2017).

In this work, we used 823 gene trees to obtain the branch-effect. The method to infer time tree is exact the same with Wu et al., 2017. The fossil calculation is also the same with Wu et al., 2017 (Table S3 in Wu et al., 2017). We deposited the input data (Fossil calculation tables, gene trees, branch-effect tree) and raw output tree in Data Availability, please have a look at them. (lines 307, 368)

Point 20: “A history of gene duplication and loss was estimated by reconciling gene trees with the species tree.”(lines 356-357) How looks the reconcilation process?

Response 20: We apologize for the unclear description. We revised this sentence to “A history of gene duplication and loss was estimated by reconciling gene trees with the species tree and plotted by R.” The code is attached to the link in Data Availability. (lines 311-312, 368)

Point 21: “A phylogenetic tree”(line 361) What kind of phylogenetic tree were constructed and how?

Response 21: We apologize for the unclear description. We constructed maximum likelihood tree. The tree construction method was described above in Materials and methods part. (lines 292-296)

Point 22: “We identified significant associations using the lasso logistic regression procedure as implemented in the glmnet package [56] in R.”(line 361) Can I see source code of this regression analysis?

Response 22: OK, we attached source code of this regression analysis in Data Availability link. (lines 354, 368)

Yellow highlights:

Point 23: (line 12) “categories” >> “group”

Response 23: We revised this sentence based on your comment above. (line 12)

Point 24: (line 16-17) “mammals experienced an upsurge in TF losses around 100 million years ago and also near the K–Pg boundary, thus implying a relationship with the divergence of placental animals.” >> “placental mammals experienced an upsurge in TF losses in relation to TF gains around 100 and 66 million years ago implying probable correspondence with increased rate of their diversification during macroevolution process.”

Response 24: Thank you for your valuable comments. We revised this sentence to “Gene tree vs. species tree reconciliation revealed that placental mammals experienced an upsurge in TF losses around 100 million years ago (Mya) and also near the Cretaceous–Paleogene boundary (K–Pg boundary, 66 Mya). Early Euarchontoglires, Laurasiatheria and marsupials appeared between 100 and 95 Mya and underwent initial diversification. The K-Pg boundary was associated with the massive loss of dinosaurs, which lead to adaptive radiation of mammals.” (lines 16-21)

Point 25: (line 20-21) “Furthermore, massive TF losses are significantly correlated four life history traits, possibly through rewiring of regulatory networks.” >> “Furthermore, the recorded results indicated that massive TF losses are significantly correlated with four distinct life history traits of mammals, possibly through restructuring of their genetic regulatory networks.”

Response 25: Thank you for your valuable comments. To provide a clear summary, we removed this sentence. (lines 21)

Point 26: (line 21-23) “We analyzed rates of molecular evolution of regulated target genes. Surprisingly, TF loss decelerated, rather than accelerated, molecular evolutionary rates of their target genes, suggesting increased functional constraints to survive target genes.” >> “The analysis of molecular evolution rates of selected target genes by us surprisingly revealed trend of deceleration in rate of TF loss, indicating enhancement of functional constraints for retainment of genes that are crucial during macroevolution of mammals.”

Response 26: Thank you for your valuable comments. We revised it to “Surprisingly, TF loss decelerated, rather than accelerated, molecular evolutionary rates of their target genes. As the rate of molecular evolution is affected by the mutation rate, the proportion of neutral mutations, and the population size, the decrease of molecular evolution may reflect in-creased functional constraints to survive target genes.”(lines 21-24)

Point 27: (lines 23-27) “As analysis covers 1,342,129 TF-TGs by chip-seq data, 9,396 by literature curations and showed consistent result: currently surviving genes have been preserved by TF loss, especially during mammal macroevolution thus promoting biodiversity.” >> “Performed by us the analysis of 1,342,129 transcription factor's target genes (TF-TGs) chip-seq data and 9,396 literature curations unambiguously showed that preservation of genes by TF loss plays important role in mammal macroevolution, promoting their presently observed high level of biodiversity.”

Response 27: Thank you for your valuable comments. To provide a concise abstract, we removed this sentence. (lines 24)

Point 28: (line 32) “taxa” >> “”

Response 28: We removed taxa in this sentences. (line 30)

Point 29: (line 33-35) “in gene expression and regulation are believed to be the major source of species phenotypic variation as well as important factors in evolution [1].” >> “are believed to be important factors in organisms evolution, being major source of among species phenotypic variation [1].”

Response 29: We revised this sentence based on your comment above. (line 31-32)

Point 30: (line 35-36) “gene regulators, perform the” >> “regulators of gene expression, perform an”

Response 30: We revised this sentence based on your comment above to explain what TF are: “Transcription factors (TFs) are sequence-specific DNA-binding trans-regulatory proteins of gene expression that perform an initial step of DNA decoding” (lines 33-34)

Point 31: (line 37) “TFs have many important functions in eukaryotes [3,4].” >> “Moreover, TFs have many other important functions in eukaryotes [3,4].”

Response 31: We revised this sentence based on your comment above.(lines 34-35)

Point 32: (line 41-42) “induced” >> “”; “cancer” >> “cancers”

Response 32: We revised this sentence based on your comments above. And the “induced pluripotent stem cells”>>” induced pluripotent stem (iPS) cells” (line 40)

Point 33: (line 44) “to varying degrees” >> “”

Response 33: We revised this sentence based on your comment above. (line 42)

Point 34: (line 45-48) “diversification” >> “biodiversification”; “simplification events during” >> “”; “history of metazoans, convergent losses of complexity in fungi, and simplification during early eukaryotic evolution [11]” >> “early evolutionary histories of metazoans, fungi and eukaryotes [11]”

Response 34: We revised this sentence based on your comments above. (lines 43-44)

Point 35: (line 48-49) “Nonadaptive simplification, such as drift, can lead to the accumulation of slightly deleterious mutations in bacteria [12]. Adaptive” >> “In contrast to nonadaptive simplification, such as drift, which can lead to the accumulation of slightly deleterious mutations in bacteria [12], adaptive”

Response 35: We revised this sentence based on your comment above.(lines 45-46)

Point 36: (line 55-56) “The” >> “Available studies shows that some of”; “even” >> “much”

Response 36: We revised this sentence based on your comments above. (lines 52-53)

Point 37: (line 58) “possibly” >> “which is”

Response 37: We revised this sentence based on your comment above. (line 55)

Point 38: (lines 58-60) “TF loss leading to major diversification has occurred in eukaryotes–for instance, the convergent simplification of adaptin complexes in flagellar apparatus diversification [11].” >> “For example, the convergent simplification of adaptin complexes in flagellar apparatus diversification caused major diversification in eukaryotes [11].”

Response 38: We revised this sentence based on your comments above. (lines 56-57)

Point 39: (lines 60-64) “As another example, the unexpectedly complex list of Wnt family signaling factors evolved in early multicellular animals about 650 million years ago (Mya). Functional and phenotypic diversification of the mouth was caused by the loss of Wnt family signaling factors during animal evolution [16].” >> “Another example is functional and phenotypic diversification of the animal mouths by the loss of signaling factors complex of the Wnt family, which evolved 650 million years ago (Mya) in the early multicellular animals [16].”

Response 39: We revised this sentence based on your comments above. (lines 57-60)

Point 40: (line 65) “How TFs work when mammalian species quickly adapt to” >> “So far, an exact mechanism of TFs work during fast adaptation of mammals to”

Response 40: We revised this sentence based on your comment above. (lines 61-62)

Point 41: (line 68) “actual” >> “real”

Response 41: We revised this sentence based on your comment above. (line 64)

Point 42: (lines 75-77) “Here” >> “In this study”; “enrichment in the macroevolutionary process. The role of TFs in macroevolutionary processes is further discussed” >> “and address the role of TFs in the macroevolutionary process of mammals”

Response 42: We revised this sentence based on your comments above. (lines 73-74)

Point 43: (line 83) “TFs were simplified during mammalian evolution” >> “The analysis of evolutionary history of Transcription Factors (TFs) in mammalians”

Response 43: We revised this sentence based on your comment above. (line 78)

Point 44: (line 84) “history of TF” >> “evolutionary history of TFs”

Response 44: We revised this sentence based on your comment above. (line 79)

Point 45: (line 89) “dates back” >> “is dated back”

Response 45: We revised this sentence based on your comment above. (line 84)

Point 46: (line 95) “Early Euarchontoglires, Laurasiatheria and marsupials” >> “Marsupialia as well as early Euarchontoglires and Laurasiatheria”

Response 46: Thank you for your valuable comment. We revised it to “Between 100 and 95 Mya, mammals, especially Placental mammals underwent initial diversification, the average TF loss rate reached its first peak” (lines 90-91)

Point 47: (line 102, Figure 1) “orthologous group” >> “TF orthologous group”

Response 47: We revised this sentence based on your comment above.(line 497, Figure 1)

Point 48: (line 111) “history;” >> “macroevolution;”

Response 48: We revised this sentence based on your comment above. (line 96)

Point 49: (line 118) “, consistent with the above results. ” >> “”

Response 49: We revised this sentence based on your comment above. (line 102)

Point 50: (line 119) “loss” >> “TF loss”

Response 50: We revised this sentence based on your comment above. (line 103)

Point 51: (lines 126-128) “Notung[21] calculates the inconsistency” >> “The applied in our study Nothung software[21] enabled for estimation of the inconsistency”; “tree but” >> “tree, ”; “inconsistency of” >> “”; “which is based on effective population size” >> “and short branches.”;

Response 51: We revised this sentence based on your comments above. (lines 111-114)

Point 52: (line 128-129) “We applied” >> “For this purpose,”; “-104, 105, 106 and 107- as” >> “(104, 105, 106 and 107) were applied”;

Response 52: We revised this sentence based on your comments above. (lines 114-115)

Point 53: (line 132) “TF events.” >> “TFs.”

Response 53: We revised this sentence based on your comment above.(line 117)

Point 54: (line 137-138, Figure 2) “showed same loss pattern.” >> “short branches.”; “red” >> “red bars”

Response 54: We revised this sentence based on your comments above. (line 506, Figure 2)

Point 55: (lines 141-143) “such as platypus data, we traced ancient mammal TFs, 982 TF orthologous groups originated before mammal common ancestor, by an additional 11 outgroups” >> “traced 982 ancient mammalian TF orthologous groups, which originated before mammal common ancestor by inclusion an additional 11 outgroups”

Response 55: We revised this sentence based on your comments above. (lines 121-123)

Point 56: (line 145-146) “amplification” >> “gain”; “and” >> “and then after this period the loss of TF started to dominate”;

Response 56: We revised this sentence based on your comments above. (lines 124-125)

Point 57: (line 150-152, Figure 3) “red” >> “red bars”; “are” >> “originated”; “groups” >> “groups of TF”;

Response 57: We revised this sentence based on your comments above. (lines 511-513, Figure 3)

Point 58: (line 167, Table 1) “associate” >> “associated”

Response 58: We revised this sentence based on your comment above. (line 530)

Point 59: (line 170) “correlated” >> “linked”

Response 59: We revised this sentence based on your comment above. (line 137)

Point 60: (line 172) “Among” >> “Thus, among”

Response 60: We revised this sentence based on your comment above. (line 139)

Point 61: (lines 175-176) “NACC1” >> “NACC1 and disruption in EPM2A gene”; “[25].

” >> “[25] [27].”

Response 61: We revised this sentence based on your comments above.(lines 142-143)

Point 62: (line 177-178) “EPM2A can also lead to epilepsy [27]. ” >> “”; “of traits around K-Pg boundary” >> “on these traits around K-Pg boundary period”

Response 62: We revised this sentence based on your comments above.(lines 144-145)

Point 63: (line 178-179) “loss helps preserve solitary for currently solitary mammal. Table 1 also shows this pattern.” >> “loss is considered to help the preservation of solitariness in extant solitary mammals that is also reflected in our results (Table 1).”

Response 63: We revised this sentence based on your comment above. (lines 146-147)

Point 64: (line 193) “temperature.” >> “that likely promoted the adaptation to a nocturnal lifestyle.”

Response 64: We revised this sentence based on your comment above. (lines 156-157)

Point 65: (line 200-201) “Insectivorous mammalian species usually have weak eyes. TF loss helps divergence of insectivory mammals and non-insectivorous.” >> “Thus, TF loss likely played important role in divergence between insectivory (poor eye-sight) and non-insectivorous (good eyesight) mammals.”

Response 65: We revised this sentence based on your comment above.(lines 163-164)

Point 66: (line 205) “has” >> “that”

Response 66: We revised this sentence based on your comment above.(line 168)

Point 67: (line 208) “fit niches.” >> “fit to specific niches.”

Response 67: We revised this sentence based on your comment above.(line 172)

Point 68: (line 211-214) “TF TGs” >> “transcription factor's target genes (TF-TGs)”; “(Figure 5). Within it,” >> “based on”; “[38]” >> “[38] (Figure 5)”

Response 68: We revised this sentence based on your comments above. (lines 175-178)

Point 69: (line 221) “conserve” >> “conserved”

Response 69: We revised this sentence based on your comment above. (line 186)

Point 70: (line 239-241) “With selected gene profiles, species be different. Possibly” >> “Likely”; “difference to creatures.” >> “inter-species differentiation.”

Response 70: We revised this sentence based on your comments above. (lines 194-195)

Point 71: (line 252-254) “As shown in Figure 1, the three branches of” >> “Our study showed that three main branches of mammals, i.e.,”; “history.” >> “history (Figure 1)”

Response 71: We revised this sentence based on your comments above. (lines 197-200)

Point 72: (line 262-266) “boundary, clusters of loss events are apparent on branches, especially the three branches leading to” >> “boundary (66 Mya), clusters of loss events are especially apparent on three branches leading to”; “both types of feeders” >> “omnivores”

Response 72: We revised this sentence based on your comments above. (lines 208-211)

Point 73: (line 280-282) “Bacterial” >> “Research on bacterial”; “mutations” >> “by mutations”; “the adaptation of bacterial populations to” >> “their adaptation”;

Response 73: We revised this sentence based on your comments above. (lines 225-227)

Point 74: (line 286) “TF loss” >> “TF loss takes place”

Response 74: We revised this sentence based on your comment above. (lines 231)

Point 75: (line 307) “mammalian TF proteins” >> “specific for mammals”

Response 75: We revised this sentence based on your comment above. (line 258)

Round 2

Reviewer 1 Report

The revised version addresses my concerns satisfactorily